# Meta-D$^2$AG: Causal Graph Learning with Interventional Dynamic Data

**Tian Gao**[*]
IBM Research

**Songtao Lu**[*]
The Chinese University of Hong Kong

**Junkyu Lee**
IBM Research

**Elliot Nelson**
Independent

**Debarun Bhattacharjya**
IBM Research

**Yue Yu**
Lehigh University

**Miao Liu**
IBM Research

## Abstract

Causal discovery in the form of a directed acyclic graph (DAG) for dynamic time series data has been widely studied in various applications. In this work, we propose a dynamic DAG discovery algorithm, Meta-D$^2$AG, based on online meta-learning. Meta-D$^2$AG is designed to learn dynamic DAG structures from potentially nonlinear and non-stationary time series datasets, accounting for changes in both parameters and graph structures. Unlike most of the existing work focusing on observational, offline, and/or stationary settings, Meta-D$^2$AG explicitly treats data collected at different time points with distribution shifts as distinct domains, which is assumed to occur as a result of external interventions. Moreover, Meta-D$^2$AG involves a new online meta-learning framework to take advantage of the temporal transition among existing domains such that it can quickly adapt to new domains with few measurements. A first-order optimization approach is utilized to efficiently solve the meta-learning framework, and theoretical analysis establishes the identifiability conditions and the convergence of the learning process. We demonstrate the promising performance of the proposed meta learning framework through better accuracy on benchmark datasets against state-of-the-art baselines.

## 1 Introduction

Probabilistic graphical models [50, 32] and causal graphical models [51] have been an active area of research in machine learning. Dynamic probabilistic graphical models [3] are particularly suitable to capture dynamics in temporal data like time series by explicitly modeling how variables change over time. These probabilistic models have been successfully applied to real-world problems, such as for neuroscience [55], molecular biology [35] and computer vision [42] tasks.

Dynamic Bayesian networks (DBNs) [12, 43] are among the most popular dynamic graphical models in the literature, initially proposed for discrete-time models with discrete variables. Later, plenty of subsequent work extended these models to consider continuous variables that are more suitable for time series data since they often involve continuous-valued measurements. For example, DBNs are used for structured vector auto-regressive (SVAR) models in the statistics and econometrics literature [56, 57, 13, 64, 34, 30, 66, 65]. Parameter learning and directed acyclic graph (DAG) structure learning from *observational time series data* are the typical learning tasks for DBNs. One can categorize standard structure learning methods into score-based [23, 8, 5], constraints-based [62, 67, 10, 40, 17, 60], or more recently, neural-based methods [41]. A more tractable continuous optimization framework has been proposed based on the algebraic characterization of the DAG [77] to address super-exponential computational complexity in exact structure learning. This

---

[*]Equal Contributions. Corresponds to:{tgao@us.ibm.com, stlu@cse.cuhk.edu.hk}.

39th Conference on Neural Information Processing Systems (NeurIPS 2025).

continuous optimization formulation [46, 70, 74] has been successfully extended to nonlinear neural models [73, 33, 26, 78, 71, 72] and dynamic DAG structure learning [49, 47, 25].

Despite its success, typical DAG learning has one major limitation. Often, the data distribution may change over time and the model parameters or even graph structures may fluctuate. For instance, in applications such as monitoring sensor networks, different sensors may be active at different times due to operating costs or weather conditions. Changes in city policies, urban development, or even major events (such as festivals or construction projects) can lead to temporal shifts in the data distribution. In retail, the sales of certain products may exhibit seasonal variations, such as increased demand for winter clothing during colder months. Standard DAG learning algorithms assume datasets involve samples that are independent and identically distributed (i.i.d.), ignoring correlations across time steps. Moreover, structure learning in DBNs over time series data generally ignores the distribution shift and assumes stationarity, which may affect the accuracy of the learned graphs. Datasets collected over a long period of time could contain distribution changes, such as monthly data or even yearly data. When these changes are significant, one may have to re-learn the graph from scratch, which loses relevant historical information and reduces learning efficiency, especially if some parameters or graph edges remain constant over time.

As a step toward addressing such non-stationary problems, we propose a meta learning based dynamic DAG learning approach, Meta-D$^2$AG, to model such distribution shifts over time. We consider the data collected over different periods as different domains under the distribution shifts, and utilize historical data to learn a set of shared parameters across the whole domain and private parameters that adapt to domain-specific changes per domain. In Meta-D$^2$AG, we view distribution shifts as caused by external interventions, e.g., updating the public health policy would influence the health record. Accordingly, we utilize the explicit intervention distribution when intervention targets are known, or the observational distribution when those targets are unknown.

The proposed framework employs a two-level optimization process to separate the invariant and domain-specific subspaces, and it can quickly adapt to a new data distribution and achieve higher graph learning accuracy with a few samples. This procedure is particularly beneficial when the data collector may not be able to capture a sufficient amount of time series data due to non-stationarity, making a low sample requirement crucial. Moreover, the chosen meta-learning framework is especially suitable for those non-stationary environments where some aspects of the environment evolve slowly or change periodically. This characteristic can make learning stationary components easier, as data across different interventions or domains are observed.

There have been a few works using a transfer-learning approach to handle invariant causal structures across domains, but they are mainly for reinforcement learning or observational time series data [11, 45, 37, 29]. There is also a line of works for multi-task DAG learning, but they require additional order information to be provided [6] and do not consider new domains in an online setting [2]. [25] considers a potential heterogeneous distribution, but it assumes a fixed graph structure where only parameters are assumed to change.

The major **contributions** of our work are as follows:

- We propose a novel meta-learning framework via bilevel optimization for time series DAG learning that explicitly models intervention as the source of distribution shift. It can handle both observational and interventional data distributions with nonlinear relationships by formulating a non-stationary collection of dynamic datasets as different domains.
- We propose an efficient first-order optimization approach to solve the proposed bilevel optimization framework with constraints in the lower level.
- Our theoretical analysis establishes the identifiability conditions in the batch setting and provides convergence rates in the online setting.

## 2 Background and Related Work

Consider a set of realizations of a potentially non-stationary time series, with each individual realization of size $T$ in the form of $\boldsymbol{X}_t := [x_{t,i}]_{i=1}^d \in \mathbb{R}^d$. Here $t \in \{0, \ldots, T\}$ represents the time index, $\boldsymbol{X}_t$ represents the observed values of all $d$ number of variables in an observational or interventional time series dataset, and $x_{t,i}$ denotes the $i$-th component of $\boldsymbol{X}_t$. For simplicity of notation, we consider one sequence $\boldsymbol{X}_t$ without loss of generality.

A **causal Bayesian network** [51] is defined by a distribution $P_X$ over a set of random variables $X \in \mathbb{R}^d$ and a DAG $\mathcal{G} = (V, E)$ with nodes $V$ and edges $E$. Each node $V_i \in V = \{V_1, ..., V_d\}$ is associated with a random variable $x_i$ and each edge $(i, j) \in E$ represents a direct causal relation from variable $x_i$ to $x_j$. With a slight abuse of notation, we will use $X$ and $V$ interchangeably. We assume the distribution $P_X$ is Markov with respect to graph $\mathcal{G}$, which enables the factorized joint distribution as $P(X) = \prod_{j=1}^{d} p_j(x_j|x_{\pi_j^G})$, where $\pi_j^G$ is the set of parents of node $j$ in the graph $\mathcal{G}$ and $x_B$ denotes the instantiations of a subset of $X$ whose indices are $B \subset V$. We also assume causal sufficiency, i.e., there are no hidden common causes between any pair of variables in $X$ [52]. As typically assumed in such models, there may exist instantaneous or contemporaneous influences as well as a time-delayed impact among variables.

Among many possible ways to model $P_X$ in a time series dataset, we follow the typical setting in [49], characterizing $X$ through a standard structured vector autoregressive (SVAR) model [13, 64, 30]:

$$\boldsymbol{X}_t = \boldsymbol{X}_t \mathbf{W}^a + \boldsymbol{X}_{t-1} \mathbf{W}_1^b + ... + \boldsymbol{X}_{t-p} \mathbf{W}_p^b + \boldsymbol{Z}_t, \tag{1}$$

where $t \in \{p, ..., T\}$ with horizon $T$, $p$ is the autoregressive order, and $\boldsymbol{Z}_t \in \mathbb{R}^d$ is a vector of noise variables drawn from any continuous distribution. We assume that $\boldsymbol{Z}_t$ is independent of $\boldsymbol{Z}_{t' \neq t}$ and of $\boldsymbol{X}_{t'}$ for all $t' < t$. The $d \times d$ matrices $\mathbf{W}^a$ and $\mathbf{W}_i^b$, $i \in \{1, ..., p\}$, represent weighted adjacency matrices for the intra-slice and inter-slice edges in $\mathcal{G}$, respectively, and they model the contemporaneous and time-lagged causal relations. Eq. 1 can be written in the matrix form as $\mathbf{X} = \mathbf{X} \mathbf{W}^a + \mathbf{Y}_1 \mathbf{W}_1^b + ... + \mathbf{Y}_p \mathbf{W}_p^b + \mathbf{Z}$, where $\mathbf{X} \in \mathbb{R}^{n \times d}$ is a matrix whose rows are $\boldsymbol{X}_t$, $\mathbf{Z} \in \mathbb{R}^{n \times d}$ is a matrix formed similarly by $\boldsymbol{Z}_t$, and $\mathbf{Y}_j$, $j \in \{1, ..., p\}$, are time lagged versions of $\mathbf{X}$. The number $n$ is the effective sample size, which is equal to $T - p + 1$.

The goal of typical **causal structure learning** tasks is to recover the DAG $\mathcal{G}$ using samples from $P_X$ and/or from the interventional distributions. We follow a continuous constrained optimization re-formulation for DAG learning [75] that uses a continuous DAG constraint, $h(\mathbf{W}) = 0$ on the weighted adjacency matrix $\mathbf{W}$ to avoid the combinatorial search on the feasible solutions $\mathbf{W}$:

$$\min_{\theta, \mathbf{W}} \mathcal{L}_\theta(\mathbf{X}; \mathbf{W}) + \lambda \Omega(\theta, \mathbf{W}) \quad \text{s.t.} \quad h(\mathbf{W}) = 0, \tag{2}$$

where $\mathcal{L}_\theta$ is the loss function and $\theta$ indicates parameters other than $\mathbf{W}$, $\Omega(\theta, \mathbf{W})$ is a regularization term on model parameters and/or the edge complexity in $\mathbf{W}$ with a tunable regularization parameter $\lambda$. [75] proposed $h(\mathbf{W}) = \text{Tr}(e^{\mathbf{W}}) - d$, where $\text{Tr}$ represents the matrix trace, and showed that the graph is acyclic if and only if the constraint $h(\mathbf{W}) = 0$. Typically, the loss $f_\theta$ can be the least square loss [75] in linear structured equation models (SEM), or evidence lower bound [73], among other losses [26]. The problem can be approximately solved using the classic augmented Lagrangian method.

For time series datasets, DYNOTEARS [49] extends the continuous optimization framework to DBNs by explicitly modeling the intra-slice and inter-slice adjacency matrices separately with a linear SEM model:

$$\min_{\theta, \mathbf{W}, \mathbf{A}} \mathcal{L}_\theta(\mathbf{X}; \mathbf{W}^a, \mathbf{W}^b) + \lambda \Omega(\theta, \mathbf{W}^a) \quad \text{s.t.} \quad h(\mathbf{W}^a) = 0, \tag{3}$$

where $\mathbf{W}^a$ is the matrix for intra-slice edges and $\mathbf{W}^b$ is the matrix for inter-slice connections, and $L_\theta(\mathbf{X}; \mathbf{W}^a, \mathbf{W}^b) = \frac{1}{n} ||\mathbf{X} - \mathbf{X} \mathbf{W}^a - \mathbf{Y} \mathbf{W}^b||_F^2$. Here $n$ is the total sample size, and $|| \cdot ||_F$ is the Frobenius norm. To distinguish with the Frobenius norm, we use $|| \cdot ||_2$ to denote the (vector) $\ell_2$-norm.

Beyond observational time series data, learning a causal graph structure in Markov Decision Process (MDP) requires additional machinery to model actions, and this can be seen as interventional dynamic DAG learning. Let a binary indicator matrix $\mathbf{R}^{\mathcal{I}} = [r_{kj}^{\mathcal{I}}] \in \{0, 1\}^{K \times d}$ encode the interventional family $\mathcal{I}$ such that $r_{kj}^{\mathcal{I}} = 1$ when $x_j$ is an intervened target in $I_k$, and 0 otherwise. For each intervention family $k$, we denote the weighted intra-slice adjacency matrix by $\mathbf{W}_{(k)}^a \in \mathbb{R}^{d \times d}$ and the weighted $p$-order inter-slice adjacency matrix by $\mathbf{W}_{(k)}^b \in \mathbb{R}^{pd \times d}$. Let $\mathbf{W}_{(k)} = \{\mathbf{W}_{(k)}^a, \mathbf{W}_{(k)}^b\}$, and $\mathbf{W} = \{\mathbf{W}_{(1)}, ..., \mathbf{W}_{(K)}\}$ represent the collection of all matrices over $K$ interventional families. Then, IDYNO [18] formulates the continuous optimization as

$$\min_{\mathbf{W}, G, \theta} \frac{1}{n} \sum_{k=1}^{K} \sum_{j=1}^{d} \left[ \mathcal{L}_{\theta_j}(\mathbf{X}, \text{MLP}(\mathbf{X}; \theta_j, \mathbf{W}_{(1)}))^{1-r_{kj}^{\mathcal{I}}} \cdot \mathcal{L}_{\theta_j}(\mathbf{X}, \text{MLP}(\mathbf{X}; \theta_j, \mathbf{W}_{(k)}))^{r_{kj}^{\mathcal{I}}} \right] \quad \text{s.t.} \quad h(\mathbf{W}) = 0$$

where $\theta_j$ is the parameter of one MLP that predicts the value of each variable $X_j$.

The above learning procedure has several limitations: 1) it needs offline data, meaning that the data needs to be saved prior to the learning process and they cannot adapt to new data when distributions change, and 2) the interventional family is fixed and limited to a known fixed number $K$, while $K$ changes over time in practice. Ultimately, the method neglects the sequential and non-stationary aspects of time series problems. This problem is also relevant in reinforcement learning, where the policies are learned and updated online, and hence the distributions and underlying MDP may change particularly around the action variables.

## 3 Meta-Learning for Dynamic DAG Learning

We focus on DAG learning for times series data, which can quickly adapt to non-stationary distributions and graphs. First, we propose a generic meta DAG learning framework via bilevel optimization for time series data when all the data are given under the batch or offline setting. Then, we extend the framework to the online setting, where new data from different domains are constantly arriving. Recall that existing work on online learning of DAG structures [9, 31] have largely focused on parameter learning settings or assumed i.i.d data. Finally, we present an efficient online Meta-D$^2$AG an algorithm based on the single-level penalty-based bilevel gradient descent.

### 3.1 Problem Formulation with Meta DAG Learning

Consider a non-stationary time series setting where data can be divided into $Q$ disjoint domains $\{1, \ldots, Q\}$. In each domain $q$, time series $\boldsymbol{X}_q = [\boldsymbol{X}_{q,t}]_{t=1}^T$ is stationary; however, one or more variables may form different distributions across the domains. The goal is to recover a set of causal DAGs, represented with weighted adjacency matrices $\boldsymbol{W}_q = \{\boldsymbol{W}_q^a, \boldsymbol{W}_q^b\}_{q=1}^Q$, governing the underlying data generation process. Given the potential for variation across domains, a desired property of effective learning algorithms is the ability to use minimal samples at each domain. This can be achieved by leveraging historical data from previous domains that share similarities with the current domain. A meta-learning approach is well-suited to meet this requirement.

**Meta Learning.** A common view of meta-learning [24] is to learn a general purpose algorithm that can generalize across a set of tasks $\mathcal{M}$ with shared parameters, and learn a new task more efficiently given the previous tasks. The goal is to evaluate the expected loss of data over a distribution over tasks $\mathcal{M}$. In general, meta-learning algorithms require a set of meta-training and meta-testing tasks drawn from the distribution $p(\mathcal{T})$. This way we can learn new task-specific parameters quickly with few samples. Meta-learning has been applied to sequential data and in particular reinforcement tasks [16, 44, 68, 54, 79], and continuous learning [59, 20].

In a standard two-step meta-learning procedure, domains $Q$ are divided into source $M$ and target $U$ datasets, $|Q| = |M| + |U|$, for meta-train and meta-test steps. The meta-training step uses a set of $M$ tasks with data $\mathcal{D}_{\text{source}} = \{(\mathcal{D}_m^{\text{train}}, \mathcal{D}_m^{\text{test}})\}_{m=1}^M$. Let $\Phi_{\text{s}}$ and $\Phi_{\text{p}}$ be the shared parameters over all tasks and the private parameters specific to each particular task, respectively. The meta-training stage aims to optimize loss function $\mathcal{L}$ by

$$\Phi_{\text{s}}^* = \arg\min_{\Phi_{\text{s}}} \log \mathcal{L}(\Phi_{\text{s}} | \mathcal{D}_{\text{source}}).$$

The meta-test stage finds the optimal parameters $\Phi_{\text{p}}$ for each specific task with data $\mathcal{D}_{\text{target}} = \{(\mathcal{D}_u^{\text{train}}, \mathcal{D}_u^{\text{test}})\}_{u=1}^U$ by

$$\Phi_{\text{p}}^* = \arg\min_{\Phi_{\text{p}}} \log \mathcal{L}(\Phi_{\text{p}}, \mathcal{D}_{\text{target}} | \Phi_{\text{s}}^*, \mathcal{D}_{\text{source}}).$$

However, our proposed setting is slightly different. The goal of each task is to learn private parameters $\Phi_{\text{p},m}$ and corresponding graph structure $\boldsymbol{W}_m$ with few samples $\mathcal{D}_{\text{source}}$ and shared parameters $\Phi_{\text{s}}$ in meta-training. One can indeed adapt to data from a new domain $u$ and infer $\Phi_{\text{p},u}$ in meta-testing, but since DAG learning is an unsupervised learning task, the meta-test stage is optional. Note that none of the existing meta-learning algorithms focuses on learning a causal graph that is applicable to general interventional time series datasets.

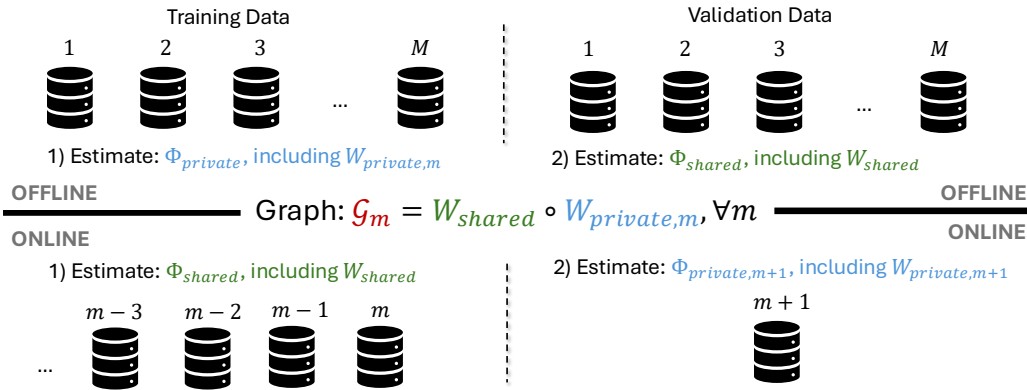

Figure 1: Overview of the proposed Meta-D2AG framework, with offline and online settings. Upper Half: In the offline setting, Meta-D$^2$AG uses different sets of data to iteratively learns the private and shared parameters of each domain to infer the causal graph $A_m$ for each domain; Lower Half: Meta-D$^2$AG can also adapt to the online setting by using data of incoming domains to infer domain-specific private graphs, given the shared parameters learned from history.

## 3.2 Meta-Learning for Dynamic DAG

We propose a novel meta-learning framework [39] for dynamic causal DAG learning in the presence of the non-stationarity of parameters and graphs by treating each segment of the time series as a different domain. We define the private parameters $\Phi_{\mathrm{p}} = \{W_p^a, W_p^b, \Theta_p\}$, and the shared parameters $\Phi_{\mathrm{s}} = \{W_s^a, W_s^b, \Theta_s\}$, where $\Theta_p$ or $\Theta_s$ include other model parameters such as linear or nonlinear SEMs. Since DAG learning is an unsupervised task, we should modify the typical meta-learning setting accordingly. The source dataset contains a set of training data and potentially a set of validation data, i.e., $\mathcal{D}_{\mathrm{source}} = \{(\boldsymbol{X}_m^{\mathrm{train}}, \boldsymbol{X}_m^{\mathrm{val}})\}_{m=1}^M$. $\boldsymbol{X}_m^{\mathrm{train}}$ is used for training all $M$ private task parameters, and $\mathcal{D}_m^{\mathrm{val}}$ is to train shared parameters $\Phi_{\mathrm{s}}$. This batch or offline setting is illustrated by the upper half of Figure 1. When desired, the target dataset contains a new dataset to learn a DAG in new domains $U$, i.e., $\mathcal{D}_{\mathrm{target}} = \{(\boldsymbol{X}_u^{\mathrm{train}})\}_{u=1}^U$. $\boldsymbol{X}_u^{\mathrm{train}}$ is then used to learn private parameters $\Phi_{\mathrm{p},u}$ in new task $u$, given shared parameters $\Phi_{\mathrm{s}}$.

As such, we formulate our *meta gradient-based dynamic DAG learning framework* (Meta-D$^2$AG) as a **bilevel optimization** problem. In meta-training, the objective with the upper level loss $\ell$ and the lower level $g$ can be written as:

$$\min_{\Phi_{\mathrm{s}}} \frac{1}{M} \sum_{m=1}^M \ell_m(\Phi_{\mathrm{s}}, \Phi_{\mathrm{p},m}^*; \boldsymbol{X}_m^{\mathrm{val}}) \tag{4a}$$

$$\text{s.t.} \; \Phi_{\mathrm{p},m}^*(\Phi_{\mathrm{s}}) = \arg\min_{\Phi_{\mathrm{p},m}} g_m(\Phi_{\mathrm{s}}, \Phi_{\mathrm{p},m}^*; \boldsymbol{X}_m^{\mathrm{train}}), \forall m, \quad \text{s.t.} \; h(\boldsymbol{W}_{\mathrm{s}}, \boldsymbol{W}_{\mathrm{p},m}) = 0 \tag{4b}$$

where $\Phi_{\mathrm{s}}$ indicates the shared parameters over different training domains and $\Phi_{\mathrm{p},m}$ indicates the private parameter evolving at each domain. $n$ is the sample size of each domain. We use a $\ell_1$-regularized least square loss for $\ell_m$ and $g_m$ in each domain, i.e., $\frac{1}{2n}\|\boldsymbol{X}_m - f_m(\Phi_{\mathrm{s}}, \Phi_{\mathrm{p},m}^*, \boldsymbol{X}_m)\|_F^2$ with either $+\eta\|\Phi_{\mathrm{s}}\|_1$ or $+\eta\|\Phi_{\mathrm{p,m}}\|_1$, where $f_m$ is the model function for each domain $m$, such as SEMs, and contains other domain-specific parameters. $\boldsymbol{X}_m$ is either the train or validation data, and we will drop it to simplify notations as $\ell_m(\Phi_{\mathrm{s}}, \Phi_{\mathrm{p},m}^*)$ and $g_m(\Phi_{\mathrm{s}}, \Phi_{\mathrm{p},m}^*)$.

In meta-testing, one can solve the inner loop to obtain a new $\Phi_{\mathrm{p},m}^*$. Recall that this bilevel formulation is different from the standard meta-learning formulation in that the lower level (or the inner loop) optimization inside the constraint contains another constraint $h(\boldsymbol{W}_{\mathrm{s}}, \boldsymbol{W}_{\mathrm{p},m}) = 0$. This poses a more complex meta-learning formulation with additional difficulties to satisfy constraints. Note that this formulation can also apply to DAG learning from i.i.d. data with multiple domains. Note also that we only enforce the acyclicity constraint $h$ at the lower level, as the upper-level constraints would be redundant.

*Remark* 3.1. The meta formulation can reduce to the vanilla continuous optimization approach [49] with a single domain, when the outer-loop optimization is not needed. Hence, the formulation is more general and covers existing continuous DAG learning frameworks.

### 3.3 Online Meta-Learning for Dynamic DAG

In dynamic datasets, domains in time series datasets may not be independent of each other, given the innate dependency overtime, even if there exists non-stationarity. Hence, we assume that new domain parameters are a function of the new time sequences $D_i$ and the parameters in the previous tasks, i.e., $\Phi_q = f(\Phi_{1,\ldots,q-1})$. Namely, all previous datasets with index $< q$ should be used to train the shared parameters for $q$ dataset for the maximal utilization of information. The source dataset contains $\mathcal{D}_{\text{source}} = \{(\mathcal{D}_i^{\text{train}}, \mathcal{D}_i^{\text{val}})\}_{i=1}^M$, with $\mathcal{D}_{<m}^{\text{train}}$ is used for training private domain-specific parameters for each domain $m$. Similarly, $D_{<m}^{\text{val}}$ is used to train shared parameters. Since the online sequence can be arbitrarily long, we can determine the total number of the historical set $M$ to train the shared parameters. On the other hand, the target dataset $\mathcal{D}_{\text{target}} = \{(\mathcal{D}_j^{\text{train}})\}_{j=1}^U$ contains $\mathcal{D}_m^{\text{train}}$ to learn private parameters in new domains given the shared parameters $\Phi_{\text{s}}$, and $U$ can be a set of sequential domains sampled directly after $M$ domains. To further refine the model in an online fashion, we could consider some $\mathcal{D}_j^{\text{val}}$ for the additional meta-training procedure to update $\Phi_{\text{s}}$. This online setting is illustrated by the buttom half of Figure 1.

As such, the modified online meta gradient-based dynamic DAG learning framework at the current domain $m$ can be written as

$$\min_{\Phi_{\text{s}}} \frac{1}{m} \sum_{i=1}^m \ell_i(\Phi_{\text{s}}, \Phi_{\text{p},i}^*) \tag{5a}$$

$$\text{s.t. } \Phi_{\text{p},i}^*(\Phi_{\text{s}}) = \arg\min_{\Phi_{\text{p},i}} g_i(\Phi_{\text{s}}, \Phi_{\text{p},i}^*), \forall i, \quad \text{s.t. } h(\boldsymbol{W}_{\text{s}}, \boldsymbol{W}_{\text{p},i}) = 0, \tag{5b}$$

**Relation between $\mathcal{G}$ and $(\boldsymbol{W}_{\text{s}}, \boldsymbol{W}_{\text{p}})$.** To instantiate the bilevel optimization framework to learn graph $\mathcal{G}$, we can choose to use a Hadamard product as $\mathcal{G}_i = \boldsymbol{W}_{\text{s}} \circ \boldsymbol{W}_{\text{p},i}$ with linear SEMs. In nonlinear SEMs with neural networks, we adopt an existing approach [78] by including $\boldsymbol{W}_{\text{s}}$ and $\boldsymbol{W}_{\text{p}}$ as the first layer parameters. Putting everything together with a two-layer feedforward neural network with domain-specific parameter $\theta_{\text{p},i}$ as an example and using the least square loss, the proposed formulation has the following forms:

$$\min_{\boldsymbol{W}_{\text{s}}} \frac{1}{m} \sum_{i=1}^m \frac{\lambda_i}{2n} \|\boldsymbol{X}_i - \theta_{\text{P},i}(\boldsymbol{W}_{\text{S}})\sigma((\boldsymbol{W}_{\text{P},i}(\boldsymbol{W}_{\text{S}}) \circ \boldsymbol{W}_{\text{S}})\boldsymbol{X}_i))\|_2^2 + \eta\|\boldsymbol{W}_{\text{S}}\|_1 \tag{6a}$$

$$\text{s.t. } \boldsymbol{W}_{\text{P},i}^*(\boldsymbol{W}_{\text{S}}), \theta_{\text{P},i}^*(\boldsymbol{W}_{\text{S}}) = \arg\min_{\boldsymbol{W}_{\text{P},i}} \frac{1}{2n}\|\boldsymbol{X}_i - \theta_{\text{P},i}\sigma((\boldsymbol{W}_{\text{P},i} \circ \boldsymbol{W}_{\text{S}})\boldsymbol{X}_i))\|_2^2, \forall i, \tag{6b}$$

$$\text{s.t. } h(\boldsymbol{W}_{\text{S}} \circ \boldsymbol{W}_{\text{P},i}) = 0 \tag{6c}$$

Interventional loss can be used when such targets are known, following equation 2.

*Remark* 3.2. The choice of multiplying shared and private parameters is inspired by existing meta-learning approaches and the masking approach in DAG learning [48]. As a result, in order to recover correct graphs in each domain, $\boldsymbol{W}_{\text{s}}$ has to include the edges of all $\boldsymbol{W}_{\text{p},i}$, hence it is unlikely to be a DAG alone. An alternative formulation $\mathcal{G}_i = \boldsymbol{W}_{\text{s}} + \boldsymbol{W}_{\text{p},i}$, which is also typical in many meta-learning literature [15], could be used, but it may be slightly more restrictive as it assumes all adjacency matrices share similar weights.

### 3.4 Penalty-based Bilevel Proximal Gradient Descent

Solving the above bilevel optimization is not straightforward, as there is an additional constraint in the inner loop. We propose a penalty-based bilevel gradient descent inspired by [61] to solve the problem. The problem can be written in a general form of

$$\min_{\Phi_{\text{s}},\{\Phi_{\text{p},i}\}} \frac{1}{m} \sum_{i=1}^m \ell_i(\Phi_s, \Phi_{\text{p},i}) \tag{7a}$$

$$\text{s.t. } \Phi_{\text{p},i} \in \mathcal{S}_i(\Phi_{\text{s}}) := \arg\min_{\Phi_{\text{p},i}'} g_i(\Phi_{\text{s}}, \Phi_{\text{p},i}'), \forall i, \quad \text{s.t. } h(\Phi_{\text{s}}, \Phi_{\text{p},i}') = 0, \tag{7b}$$

where $\mathcal{S}_i(\Phi_{\text{s}})$ denotes the optimal solution set of the $i$th lower-level problem.

Towards this end, by moving both the objectives and constraints in the inner loop as different penalty terms to the outer loop, we can write the penalized objective as:

$$\mathcal{L}(\Phi_{\mathrm{s}}, \{\Phi_{\mathrm{p},i}\}; \gamma_i, \nu_i) = \frac{1}{m} \sum_{i=1}^{m} \ell_i(\Phi_{\mathrm{s}}, \Phi_{\mathrm{p},i}) + \gamma_i\big(g_i(\Phi_{\mathrm{s}}, \Phi_{\mathrm{p},i}) - g_i^*(\Phi_{\mathrm{s}})\big) + \nu_i h(\Phi_{\mathrm{s}}, \Phi_{\mathrm{p},i}), \quad (8)$$

where $\{\gamma_i, \nu_i\}$ and $g_i^*(\Phi_{\mathrm{s}}) = \min_{\Phi_{\mathrm{p},i}} g_i(\Phi_{\mathrm{s}}, \Phi_{\mathrm{p},i})$ are penalty coefficients. The second term of equation 8 penalizes the constraints directly in the objective. One issue with equation 7a is that $m$ can be large and computation would become expensive for long-history domains. As in typical online algorithms, it is prudent to use a sliding window to capture the historical influence as follows.

$$\min_{\Phi_{\mathrm{s}}, \{\Phi_{\mathrm{p},i}\}} \frac{1}{w} \sum_{i=m-w+1}^{m} \ell_i(\Phi_{\mathrm{s}}, \Phi_{\mathrm{p},i}) \tag{9a}$$

$$\text{s.t.} \ \Phi_{\mathrm{p},i} \in \mathcal{S}_i(\Phi_{\mathrm{s}}) := \arg\min_{\Phi_{\mathrm{p},i}'} g_i(\Phi_{\mathrm{s}}, \Phi_{\mathrm{p},i}'), \forall i, \quad \text{s.t.} \quad h(\Phi_{\mathrm{s}}, \Phi_{\mathrm{p},i}') = 0, \tag{9b}$$

where $w$ denotes the size of the sliding window.

**Sliding Window Formulation.** Due to the inherent nonlinearity of neural networks, the corresponding objective functions exhibit non-convex properties. [22] has shown that it is possible to measure the regret using a local regret metric, allowing us to establish a regret bound based on a smooth nonconvex objective function. Then, the penalty-based sliding average is:

$$S_{i,w}(\Phi_{\mathrm{s}}, \{\Phi_{\mathrm{p},i}\}) = \frac{1}{w} \sum_{i=m-w+1}^{m} \ell_i(\Phi_{\mathrm{s}}, \Phi_{\mathrm{p},i}) + \gamma_i\big(g_i(\Phi_{\mathrm{s}}, \Phi_{\mathrm{p},i}) - g_i^*(\Phi_{\mathrm{s}})\big) + \nu_i h(\Phi_{\mathrm{s}}, \Phi_{\mathrm{p},i}). \tag{10}$$

As such, the optimization becomes a single level and we can adopt a proximal gradient descent method with a penalty function to optimize $\Phi = (\Phi_{\mathrm{s}}, \Phi_{\mathrm{p},i})$ together. For a more detailed discussion of the algorithm and proofs on the following convergence property, we refer to Appendix A.1.

---

**Algorithm 1** Online Meta-D$^2$AG Algorithm

---

**Require:** Data $\boldsymbol{X}_m, m \in \{1, \ldots, M\}, w, \eta, \gamma$
1: **Output**: Learned shared weighted adjacency matrix $\boldsymbol{W}_{\mathrm{s}}$ and private $\boldsymbol{W}_{\mathrm{p},m}, \forall m$
2: initialize $\boldsymbol{W}_{\mathrm{s}}$ and $\boldsymbol{W}_{\mathrm{p},0}$
3: **for** each domain $m$ **do**
4:      Initialize $\Phi_{\mathrm{p},m} := \Phi_{\mathrm{p},m-1}$
5:      **while** equation 11 holds **do**
6:          $g_i^*(\Phi_{\mathrm{s}}) = \min_{\Phi_{\mathrm{p},i}} g_i(\Phi_{\mathrm{s}}, \Phi_{\mathrm{p},i})$ {Minimize inner loop}
7:          Compute $S_{i,w}$ with equation 10
8:          $\Phi \leftarrow \mathrm{prox}_{\eta f}(\Phi - \eta \nabla_\Phi \widehat{S}_{i,w}(\Phi))$
9:      **end while**
10: **end for**

---

As there is a non-smooth term in the objective, we adopt the proximal residual, i.e., $\mathcal{P}_\eta^f(x, d) = \eta^{-1}(x - \mathrm{prox}_{\eta f}(x - \eta d)), \forall \eta > 0$ [21] in the optimality gap to quantify the local regret, where $\mathrm{prox}_{\eta f}(v) := \min_x \eta f(x) + \frac{1}{2}\|x - v\|_2^2$ denotes the proximal operator, and $d$ is any vector, $f(x) = \|x\|_1$ and $\mathrm{prox}_{\eta f}(v)$ is the soft-thresholding operator. To summarize, we show the online algorithm in Algorithm 1, where $\nabla_\Phi \widehat{S}_{i,w}(\Phi)$ denotes an approximate evaluation of $\nabla_\Phi S_{i,w}(\Phi)$ obtained by optimizing $\min_{\Phi_{\mathrm{p},i}} g_i(\Phi_{\mathrm{s}}, \Phi_{\mathrm{p},i})$ using any optimization algorithm practically, rather than evaluating it exactly at $\Phi_{\mathrm{p},i} \in \mathcal{S}_i(\Phi_{\mathrm{s}})$.

## 4 Theoretical Results

**Convergence** Before presenting our theoretical convergence results, we first introduce a set of mathematical assumptions regarding the problem.

**Assumptions 1.** We assume that $g(,)$, $h(,)$, and $\ell_i(,)$ for all $i$ are differentiable and Lipschitz continuous. Moreover, for every $i$, $g_i(\Phi_{\mathrm{s}}, \cdot)$ is gradient Lipschitz continuous with constant $L_g$ and satisfies the proximal Polyak-Łojasiewicz (PŁ) error bound.

Note that these assumptions are standard and quite moderate. For example, in the applications of DAG learning, the loss functions are typically score functions, which are smooth. In addition, the causal structure constraint $h(,)$ for DAG is also smooth. Since the non-smooth component involves the $\ell_1$-norm, it effectively penalizes variable magnitudes, which in turn ensures the boundedness of the iterates. This observation consequently implies the Lipschitz continuity of the loss functions.

The assumption of the proximal PŁ error bound [27] is primarily employed here to make the optimization process of the lower-level (or inner loop) more tractable, a condition that holds for a broad spectrum of non-convex loss functions. Specifically, when the neural network size is large, i.e., overparameterized, the associated loss function satisfies the PŁ condition. In this context, due to the presence of the non-smooth term, we utilize the proximal PŁ error bound. Under this condition, a few steps of gradient descent can approximately find the global optimal solution, meaning that the last iterates of the gradient descent can be used to approximate $g_i^*()$, allowing $S_{i,w}(\Phi)$ to be easily evaluated, denoted as $\widehat{S}_{i,w}(\Phi)$, where $\Phi$ is the concatenation of all the model parameters. This approach is both practical and can be rigorously quantified theoretically.

The convergence criterion in Line 5 of Algorithm 1 is based on the sum of gradients in the sliding window:
$$\|\mathcal{P}_\eta^f(\Phi_{i+1}, \nabla\widehat{S}_{i,w}(\Phi_{i+1}))\| > \delta/w, \tag{11}$$
where $\delta$ is a constant, which means that if the proximal residual is not sufficiently small, we continue optimizing the model parameters using the current domain data.

**Theorem 4.1.** ***Regret Bound.*** *Under these assumptions, suppose that* $\Phi := \{\Phi_s, \Phi_{p,i} \mid i = m - w + 1, \ldots, m\}$ *is generated by the algorithm. Then, the following holds:*

$$\mathrm{Reg}_w(m) = \sum_{i=1}^m \|\mathcal{P}_\eta^f(\Phi_i, \nabla S_{i,w}(\Phi_i))\|^2 = \mathcal{O}\left(\frac{m}{w^2}\right). \tag{12}$$

*Remark* 4.2. Since there are two levels in the optimization process, quantifying convergence requires solving the lower-level problems, unlike classic single-level problems. Thus, our proof must account for errors in solving both levels. Due to the nonconvexity of the loss function, we use the sliding window concept to demonstrate the stationarity of the solutions obtained.

The above regret indicates that the changes in the sliding-window task-averaged functions converge to a small value when the sliding window size is large, meaning that $\mathbb{E}_i\|\mathcal{P}_\eta^f(\Phi_i, \nabla S_{i,w}(\Phi_i))\|^2 \leq \mathrm{Reg}_w(m)/m = \mathcal{O}(w^2)$. It is worth noting that this algorithm achieves the optimal regret bound, as equation 12 matches the optimal regret of the time-smoothed online proximal gradient method for solving classic single-level nonconvex minimization problems [22, 21].

**Other Assumptions** We also make a few standard assumptions for dynamic DAG learning [49], such as causal sufficiency (no latent variables), following an SEM model (linear or nonlinear), and sampling frequency of the event sequence is at least as high as the fluctuations in the underlying causal process. However, we relax the typical assumption that the graph structure is fixed over time, and assume that it is only fixed within certain time steps (as fixed within a domain in the meta formulation).

**Identifiability** In Appendix A.2, we demonstrate that, under a mild condition on the shared parameters $W_s$ estimated from previous domains, the earlier identifiability results from [4, 18] can be extended to our method for learning the graph with adjacency matrix $W_s \circ W_{p,m}$ within a given domain $m$.

**Complexity** Our general meta-learning framework has the same computation complexity, with respect to the number of nodes, as existing continuous optimization approaches. The additional computation in our formulation mainly comes from the lower-level optimization, where we need to learn a DAG for each of $M$ domains. The upper level can be solved faster without any constraints.

## 5 Empirical Evaluation

We evaluate our proposed meta dynamic DAG learning algorithm against some of the existing DAG learning algorithms, designed for dynamic time series datasets and/or with multi-domain capabilities.

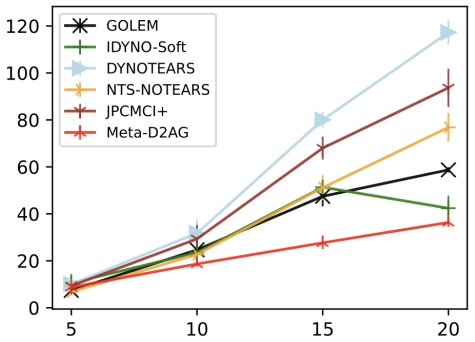
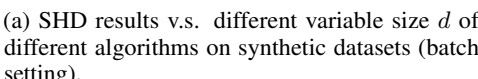
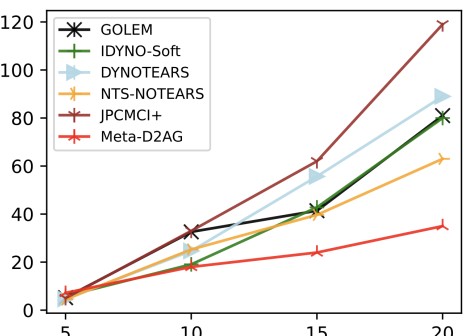

(a) SHD results v.s. different variable size $d$ of different algorithms on synthetic datasets (batch setting).

(b) SHD results v.s. different variable size $d$ of different algorithms on synthetic datasets (online setting).

We test the algorithms on synthetic datasets as well as a simulated reinforcement learning (RL) environment, Sprites World, with intervention datasets. The experiments are run on a machine with a 3.2 GHz CPU and 64 GB of memory.

**Baselines** We compare our method with the following baselines with available code: continuous DAG learning methods GOLEM [47], dynamic DAG learning algorithm DYNOTEARS [49], NTS-NOTEARS [63], offline interventional dynamic DAG learning algorithms IDYNO [18], and multi-domain dynamic DAG algorithm J-PCMCI+ [19]. We repeat each experiment 5 times and compare the structural Hamming distance (SHD) between the estimated graph and the ground truth graph (the lower, the better). All baselines except J-PCMCI+ are designed to handle one domain, so we learn each domain separately and report the average SHD with standard deviation. J-PCMCI+ only finds one unique graph across all domains so we use it as the graph in each domain.

## 5.1 Synthetic Datasets

We first evaluate different approaches on synthetic time series datasets. We simulate the data according to the SEM from equation 1, mostly following the setup and code from [49] for ease of comparison. The generating process consists of the following three steps: 1) Generating weighted graphs in the form $\mathbf{W^a}$ and $\mathbf{W^b}$. Directed graphs are generated first as a lower triangular matrix, and then randomly permuted, 2) Generating data matrices $\mathbf{X}$ and corresponding $\mathbf{Y}$ per $W^a$ and $W^b$, and 3) Generating interventional data with random targets and different distributions. For a more detailed generation and training process, we refer readers to Appendix B.2.

We show results for the batch setting where all 10 domains are learned jointly per equation 4, and for the online setting where we use a sliding window size of 3 previous time domains to meta-train and the current domain as the meta-test dataset. For the batch setting, since all domains are used in meta-training and there is no new meta-test data, we report the SHD of training accuracy of each domain. For the online setting, we report SHD from the meta-test domain at each time step, given the previous 3 domains as meta-training (padding in the beginning with the data from the first domain). For baseline methods in the online setting, to distinguish from the batch setting where each domain are learned separately, we feed all domain data up to the current time stamp to each method and produce a graph.

As one can see from Figure 2a, in the batch setting, our method (colored in red) has the lowest SHD. Since each domain has different graphs, it hurts the performance of J-PCMCI+. For the online setting shown in Figure 2b, Meta-D$^2$AGs again superior to baselines. In an appendix, we also conduct an ablation study with two variants of our method, one only optimizing the shared parameters and one with a linear loss function, showing the advantage of the proposed formulation. Moreover, in Meta-D$^2$AG the DAG constraint follows the matrix exponential form as used previously by [49]. Other and better DAG constraints can also be used, such as [1, 76].

**Ablation Study** In the appendix, we have included further results of many ablation studies of our approach, including the necessity of the lower level optimization on private parameters and the impact

of sequence lengths and variable variance. Overall, the proposed Meta-D$^2$AG shows promising performance in these settings.

## 5.2 Sprites World RL Environment

Sprites world [69] is a Python-based RL environment in which objects of various shapes interact with each other in 2-dimensional space. In the experiment, we created 5 and 10 objects, each associated with a continuous variable that assigns values following a linear structural equation model or non-linear structural equation model similar to causal discovery benchmarks [28, 53]. For further details about the data generation and training process, we refer readers to Appendix B.3.

As shown in Table 1, comparing with the baselines, our method achieves the lowest SHD performance for 5 and 10 objects, again demonstrating promising graph discovery performance in an RL environment.

Table 1: SHD performance of different methods in Sprites World.

| Method | $O = 5$ | $O = 10$ |
|---|---|---|
| GOLEM | $6.0 \pm 1.3$ | $14 \pm 2.4$ |
| IDYNO | $5.8 \pm 1.7$ | $12 \pm 3.0$ |
| DYNOTEAR | $11 \pm 2.3$ | $39 \pm 1.3$ |
| JPCMCI+ | $12 \pm 1.7$ | $31 \pm 0.6$ |
| Meta-D$^2$AG | $\mathbf{5.2 \pm 0.6}$ | $\mathbf{9.1 \pm 1.5}$ |

## 6 Conclusion

We propose a new Meta-D$^2$AG method for causal graph discovery in dynamic datasets. Leveraging the meta-learning framework to capture temporal transitions between different time domains is well-suited for causal graph discovery in dynamic datasets. We make three major contributions. First, we formulate a meta-learning framework tailored for dynamic DAG learning, accommodating slightly varied distributions in different domains over time. We further extend the framework to an online setting, where a windowed formulation is proposed to improve learning efficiency in scenarios with extensive historical data. Second, recognizing the challenges posed by acyclicity constraints in inner-level optimization, we propose an efficient first-order method to solve the bilevel framework. Finally, theoretical analysis and experimental evaluations are provided to show the convergence of the learning process and the superiority in empirical performances. Future research could explore methods for the automatic detection of potential distribution shifts without the need for such a priori information. For example, distribution shift detection is an active area of research, and we can adopt various strategies, such as measuring prediction error, using distribution distance, and/or hypothesis testing to detect shifts [38] and form a new domain. It would also be interesting to combine the proposed approach with irregular [7] or sub-sampled time series [36].

**Limitation and Impact Statement** One limitation of our approach is its reliance on predefined domains in time series datasets, which may be restrictive when distribution shifts occur irregularly. Obtaining such partitions of temporal data when no domain labels are provided would be a useful future direction. This paper presents work whose goal is to advance the field of causal discovery algorithms. Although there are no substantial potential societal consequences of our work, misuse could result in wrong conclusions about the causal relationships, which need to be carefully empirically validated in a controlled environment.

## Acknowledgments and Disclosure of Funding

The authors thank anonymous reviewers for their constructive feedback. The work of Songtao Lu is supported in part by the Chinese University of Hong Kong (CUHK) Direct Research Grant (Project No. 4055259). Yue Yu is partially supported by the National Science Foundation (NSF) under award DMS-2436624 and the Defense Advanced Research Projects Agency (DARPA) under award HR00112590032. This research was, in part, funded by the U.S. Government by an agreement with Cornell University. The views and conclusions contained in this document are those of the authors and do not necessarily reflect the views of Cornell University, and should not be interpreted as representing the official policies, either expressed or implied of the U.S. Government.

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

## A  Technical Appendix

### A.1  Algorithm and Convergence

As the optimization algorithm is general, for ease of reading, we simplify the notation here. Instead of using $\Phi_\mathrm{s}$ and $\Phi_\mathrm{p}$, we use $(x, y)$.

Let define
$$\ell_i(x, y_i) = f_i(\boldsymbol{X}_i^{\mathrm{val}}, \boldsymbol{W}_\mathrm{s}, \boldsymbol{W}_{\mathrm{p},i}^*(\boldsymbol{W}_\mathrm{s}), \theta_{\mathrm{p},i}^*(\boldsymbol{W}_\mathrm{s})) + \eta \|\boldsymbol{W}_\mathrm{s}\|_1 \tag{13}$$
and
$$g_i(x, y_i) = \frac{1}{2n} \|\boldsymbol{X}_i^{\mathrm{train}} - \theta_{\mathrm{p},i} \sigma((\boldsymbol{W}_{\mathrm{p},i} \circ \boldsymbol{W}_\mathrm{s}) \boldsymbol{X}_i^{\mathrm{train}}))\|_2^2 + \eta \|\boldsymbol{W}_{\mathrm{p},i}\|_1. \tag{14}$$

Then, the problem can be concisely written as
$$\min_{x, y_i} \frac{1}{m} \sum_{i=1}^m \ell_i(x, y_i) + \eta f(x) \tag{15a}$$
$$\text{s.t. } y_i \in \mathcal{S}(x) := \arg\min_{y_i} g_i(x, y_i) + \eta f(y_i) \quad \text{s.t.} \quad h(x, y_i) = 0 \tag{15b}$$

where in this case $f() := \|\cdot\|_1$, but it can generally represent any projection-friendly non-smooth term.

To solve this problem, we can apply the penalized method. Towards this end, we first write the penalized objective as follows
$$\mathcal{L}_i(x, \{y_i\}; \gamma_i, \nu_i) = \frac{1}{m} \left( \sum_{i=1}^m \ell_i(x, y_i) \right) + \gamma \left( g_i(x, y_i) - g_i^*(x) \right) + \nu h(x, y_i) + \eta f(x) + \eta f(y_i) \tag{16}$$

where $\{\gamma, \nu\}$ are penalizers, and $g_i^*(x) = \min_y g_i(x, y)$.

Recall the definition of the proximal operator:
$$\mathrm{prox}_{\eta f}(x - \eta d)$$
$$= \arg\min_z \eta f(z) + \frac{1}{2} \|x - \eta d - z\|^2 \tag{17}$$
$$= \arg\min_z \eta f(z) + \eta \langle d, z - x \rangle + \frac{1}{2} \|z - x\|^2. \tag{18}$$

Then, we can have the proximal residual defined as
$$\mathcal{P}_\eta^f(x, d) = \frac{1}{\eta} (x - \mathrm{prox}_{\eta f}(x - \eta d)), \tag{19}$$

which will serve as the optimality gap for measuring the convergence of the algorithm.

For simplicity of the notations, define $z = (x, \{y_j, j = i - w + 1, \ldots, i\})$. Then, we can construct the window average as follows
$$S_{i,w}(z) = \frac{1}{w} \sum_{j=i-w+1}^i \ell_j(x, y_j) + \gamma \left( g_j(x, y_j) - g_j^*(x) \right) + \nu h_j(x, y_j). \tag{20}$$

Due to the non-convexity and non-smoothness of the loss function, we follow the existing notation of the local regret of a policy $z_i$ up to time $m$ with window length $w$, which is defined
$$\mathrm{Reg}_w(m) = \sum_{i=1}^m \|\mathcal{P}_\eta^f(z_i, \nabla S_{i,w}(z_i))\|^2. \tag{21}$$

From the definition of $S_{i,w}(z)$, we can have
$$S_{i,w}(z) = S_{i-1,w}(z) + \frac{1}{w}(\ell_i(x, y_i) - \ell_{i-w}(x, y_{i-w}))$$
$$+ \frac{\gamma}{w} \left( g_i(x, y_i) - g_i^*(x) - (g_{i-w}(x, y_{i-w}) - g_{i-w}^*(x)) \right)$$
$$+ \frac{\nu}{w} \left( h_i(x, y_i) - h_{i-w}(x, y_{i-w}) \right). \tag{22}$$

However, due to the bilevel structure of the problem, we can only achieve an estimate of $g_i^*(x)$ through the inner loop update $\widehat{y}$, which is denoted as $\widehat{g}_i(x) = g(x, \widehat{y})$. Then, we can use $|\mathcal{P}_\eta^f(z_{i+1}, \nabla S_{i,w}(z_{i+1}))\|$ to determine whether we need to update the model.

Recall our algorithm: when $\|\mathcal{P}_\eta^f(z_{i+1}, \widehat{\nabla} S_{i,w}(z_{i+1}))\| > \delta/w$, we implement the proximal gradient descent. Otherwise, we update our model, where

$$
\begin{aligned}
\widehat{S}_{i,w}(z) = {} & S_{i-1,w}(z) + \frac{1}{w}(\ell_i(x, y_i) - \ell_{i-w}(x, y_{i-w})) \\
& + \frac{\gamma}{w}\left(g_i(x, y_i) - \widehat{g}_i(x) - (g_{i-w}(x, y_{i-w}) - \widehat{g}_{i-w}(x))\right) \\
& + \frac{\nu}{w}\left(h_i(x, y_i) - h_{i-w}(x, y_{i-w})\right).
\end{aligned}
\tag{23}
$$

Next, we will find an upper bound of $\|\mathcal{P}_\eta^f(z_{i+1}, \nabla S_{i,w}(z_{i+1}))\|$. First, we can have

$$
\begin{aligned}
& \|\mathcal{P}_\eta^f(z_i, \nabla S_i(z_i))\| \\
& \overset{(a)}{\leq} \|\mathcal{P}_\eta^f(z_i, \nabla S_i(z_i))\| + \|\mathcal{P}_\eta^f(z_i; \nabla S_i(z_i)) - \mathcal{P}_\eta^f(z_i; \widehat{\nabla} S_i(z_i))\| \tag{24} \\
& \leq \|\mathcal{P}_\eta^f(z_i, \nabla S_{i-1}(z_i))\| \\
& \quad + \frac{\gamma}{w}\left\|\nabla g_i(x_i, y_i) - \nabla g_i^*(x_i) - (\nabla g_{i-w}(x_i, y_{i-w}) - \nabla g_{i-w}^*(x_i))\right\| \\
& \quad + \frac{\nu}{w}\|\nabla h_i(x_i, y_i) - \nabla h_{i-w}(x_i, y_{i-w})\| + \frac{1}{w}\|\nabla \ell_i(x_i, y_i) - \nabla \ell_{i-w}(x_i, y_{i-w})\| \\
& \quad + \frac{\gamma}{w}\|\nabla g_i^*(x_i) - \nabla \widehat{g}_i(x_i)\| + \frac{\gamma}{w}\|\nabla g_{i-w}^*(x_i) - \nabla \widehat{g}_{i-w}(x_i)\| \tag{25} \\
& \overset{(b)}{\leq} \frac{\delta}{w} + \frac{\gamma}{w}\|\nabla g_i^*(x_i) - \nabla \widehat{g}_i(x_i)\| + \frac{\gamma}{w}\|\nabla g_{i-w}^*(x_i) - \nabla \widehat{g}_{i-w}(x_i)\| + V_{i,w} \tag{26} \\
& \overset{(c)}{\leq} \frac{\delta}{w} + \frac{\gamma}{w}L_g d_{\mathcal{S}(x_i)}(\widehat{y}_i) + \frac{\gamma}{w}L_g d_{\mathcal{S}(x_i)}(\widehat{y}_{i-w}) + V_{i,w} \tag{27}
\end{aligned}
$$

where $(a)$ is true due to the triangle inequality, $(b)$ holds by following the condition of updating the model and $V_{i,w}$ is defined as

$$
\begin{aligned}
V_{i,w} = {} & \frac{\gamma}{w}\left\|\nabla g_i(x_i, y_i) - \nabla g_i^*(x_i) - (\nabla g_{i-w}(x_i, y_{i-w}) - \nabla g_{i-w}^*(x_i))\right\| \\
& + \frac{\nu}{w}\|\nabla h_i(x_i, y_i) - \nabla h_{i-w}(x_i, y_{i-w})\| + \frac{1}{w}\|\nabla \ell_i(x_i, y_i) - \nabla \ell_{i-w}(x_i, y_{i-w})\| \tag{28}
\end{aligned}
$$

and in $(c)$ we apply the gradient Lipschitz continuity of $g_i(x, \cdot), \forall i$ and $d_{\mathcal{S}}(y) = \arg\min_{y' \in \mathcal{S}}\|y' - y\|$.

When the lower level problem satisfies the proximal PŁ error bound, i.e.,

$$
\frac{1}{\eta}\left\|y - \text{prox}_{\eta f}(y - \eta \nabla_y g(x, y))\right\| \geq c d_{\mathcal{S}(x)}(y) \tag{29}
$$

where $c$ is a constant and $y$ here is generic, representing any $y_i$, then according to [27, Appendix G] we have the function also satisfies the proximal PŁ condition, which is

$$
\frac{1}{2}\mathcal{D}_f(y; x) \geq \mu\left(F(x, y) - F^*(x)\right) \tag{30}
$$

where $F(x, y) := g(x, y) + \eta f(x), F^*(x) := g^*(x) + \eta f(x)$ and

$$
\mathcal{D}_f(y; x) = -2L_g \min_{y'}\langle \nabla g(x, y), y' - y\rangle + \frac{L_g}{2}\|y' - y\|^2 + \eta f(y') - \eta f(y). \tag{31}
$$

By the Lipschitz gradient of $g(x, y)$, we have

$$
\begin{aligned}
& g(x, y_{t+1}') + \eta f(y_{t+1}') \\
& \leq g(x, y_t') + \eta f(y_t') + \langle \nabla_y g(x, y_t'), y_{t+1}' - y_t'\rangle + \frac{L_g}{2}\|y_{t+1}' - y_t'\|^2 + \eta f(y_{t+1}') - \eta f(y_t') \tag{32} \\
& = g(x, y_t') + f(y_t') - 2L_g \mathcal{D}_f(y_t'; x) \tag{33}
\end{aligned}
$$

where $y'_t$ denotes the iterates in the inner loop.

Substituting equation 30 to equation 33 gives

$$F(x, y'_{t+1}) - F^*(x) \leq \left(1 - \frac{L_g}{\mu}\right)(F(x, y'_t) - F^*(x)) \leq \left(1 - \frac{L_g}{\mu}\right)^T (F(x, y'_1) - F^*(x)) \quad (34)$$

where $T$ denotes the number of iterations in the inner loop.

From [14, Corollary 3.6], we know that the proximal PŁ error bound implies the quadratic growth property, meaning that there exists a constant $\bar{\mu}$ such that

$$d^2_{\mathcal{S}(x)}(\widehat{y}_i) \leq \frac{\bar{\mu}}{2}\left(1 - \frac{L_g}{\mu}\right)^T (F(x, y'_1) - F^*(x)) \quad (35)$$

where $\widehat{y}_i := y'_{T+1}$. As $x$ is bounded, the term $d_{\mathcal{S}(x_i)}(\widehat{y}_i)$ shrinks to 0 at a linear rate.

Summing over $i = 1, \ldots, m$ gives

$$\begin{aligned}
&\mathrm{Reg}_w(m) \\
&= \sum_{i=1}^{m} \|\mathcal{P}^f_\eta(z_i, \nabla S_{i,w}(z_i))\|^2 \quad (36) \\
&\leq \frac{4m\delta^2}{w^2} + 4L_g^2\gamma^2 \sum_{i=1}^{m} d^2_{\mathcal{S}(x_i)}(\widehat{y}_i) + 4L_g^2\gamma^2 \sum_{i=1}^{m} d^2_{\mathcal{S}(x_i)}(\widehat{y}_{i-w}) + \frac{4}{w^2} \sum_{i=1}^{m} V_{i,w}. \quad (37)
\end{aligned}$$

When we choose $T \geq -\log_{(1-L_g/\mu)} \bar{\mu}(F(x, y'_1) - F^*(x))w^2$, we have $\sum_{i=1}^{m} d^2_{\mathcal{S}(x_i)}(\widehat{y}_i) \sim m/w^2$. The last term $\sum_{i=1}^{m} V_{i,w}$ is also referred to as the sliding window variation. When the gradients of the loss function are bounded, we automatically have that this term is $\mathcal{O}(m)$. Therefore, we can conclude that our algorithm still can achieve the optimal regret bound as $\mathcal{O}(m/w^2)$.

### A.2 Identifiability

In this section, we demonstrate that Theorem 3.2 from [18] (an extension of Theorem 1 from [4]) can be applied in our online meta-learning setting. We denote the current domain with index $m$, and recall that a model estimate $\boldsymbol{W}_s$ of the shared, cross-domain part of the graph adjacency matrix can be formed from the data from the previous $m - 1$ domains. When encountering domain $m$, our objective is to optimize the part of the adjacency matrix specific to the domain, $\boldsymbol{W}_{p,m,k=1}$ (where $k = 1$ indicates the observational graph, and $k > 1$ indexes interventions on this graph, as in [18]).

We denote with $\mathcal{G}_m$ the DAG whose edges are given by the nonzero elements of the adjacency matrix $\boldsymbol{W}^{\mathcal{G}_m} := \boldsymbol{W}_s \circ \boldsymbol{W}_{p,m,k=1}$. Our score function for $\mathcal{G}_m$ can be written (following the notation of [4])

$$\mathcal{S}_{\mathcal{I}^\star}(\mathcal{G}_m) := \sup_{\theta_m, \boldsymbol{W}|\mathcal{G}_m} \sum_{k=1}^{K} \mathbb{E}_{\boldsymbol{X} \sim p_m^{(k)}} \log f^{(k)}(\boldsymbol{X}; \boldsymbol{W}^{\mathcal{G}_m}, R^{\mathcal{I}^\star}, \theta_{m,k}) - \eta|\mathcal{G}_m| \quad (38)$$

where the regularization term $|\mathcal{G}_m| = \|\boldsymbol{W}_s \circ \boldsymbol{W}_{p,m,1}\|_0$ is the L0 norm (which counts the number of edges in $\mathcal{G}_m$), $\theta_m = \{\theta_{m,k}\}_{k=1}^{K}$ with $\theta_{m,k} = \{\theta_{m,k,j}\}_{j=1}^{d}$ are neural network parameters, $p_m^{(k)}$ is the ground truth generating distribution of the data for domain $m$ with intervention $I_k \in \mathcal{I}$, and $R^{\mathcal{I}} = \{r^{\mathcal{I}}_{kj}\}_{k,j=1}^{K,d}$ specifies the set of interventions, which influence the log-likelihood function as follows,

$$\log f^{(k)}(\boldsymbol{X}; \boldsymbol{W}, R^{\mathcal{I}}, \theta_{m,k}) = \sum_{j=1}^{d} r^{\mathcal{I}}_{kj} \cdot \log \tilde{f}(\boldsymbol{X}, \mathrm{NN}(\boldsymbol{X}, \theta_{m,k,j}, \boldsymbol{W}_k)) + (1 - r^{\mathcal{I}}_{kj}) \cdot \log \tilde{f}(\boldsymbol{X}, \mathrm{NN}(\boldsymbol{X}, \theta_{m,1,j}, \boldsymbol{W}_k))$$

where we again denote the observational distributions with $k = 1$, and sum over the model likelihood function $\tilde{f}$ (for intervention $k$) for each node $j$ in the graph. Here, "NN" denotes a neural network function, and $\boldsymbol{W}_k^{\mathcal{G}_m} := \boldsymbol{W}_s \circ \boldsymbol{W}_{p,m,k}$ is the full adjacency matrix for domain $m$ and intervention $k$ (we assume $\boldsymbol{W}_s$ does not depend on the intervention $k$). Lastly, when taking the supremum in equation 38, we use $\boldsymbol{W}|\mathcal{G}_m$ to indicate any values of the continuous matrix $\boldsymbol{W}$, subject to the constraint that edges absent in $\mathcal{G}_m$ correspond to zero entries in $\boldsymbol{W}$.

The only difference between this setting and the setting of [18] is that instead of learning an adjacency matrix, we are learning a factor $W_{\mathrm{p},m,k}$ of the full adjacency matrix $W_{\mathrm{s}} \circ W_{\mathrm{p},m,k}$. Thus, in the case where $W_S$ is set to the all-ones matrix, we recover the setting of [18] exactly. Theorem 3.2 from [18] (as well as Theorem 1 from [4]) involve the set of DAGs maximizing a score function $\mathcal{S}_{\mathcal{I}^\star}(\mathcal{G})$, where $\mathcal{I}^\star$ is the ground truth set of interventions. Our score function, equation 38, is equivalent when $W_S$ is set to the all-ones matrix.

For any given adjacency matrix $W^\star$ maximizing equation 38 (for a fixed graph $\mathcal{G}_m$), as long as the elements of $W_S$ are nonzero wherever $W^\star$ is nonzero (wherever the graph $\mathcal{G}_m$ has an edge), then there exists a matrix $W_{\mathrm{p}}$ satisfying $W^\star = W_{\mathrm{s}} \circ W_{\mathrm{p}}^\star$, for which each element is determined uniquely as $(W_{\mathrm{p}}^\star)_{ij} = (W^\star)_{ij}/(W_{\mathrm{s}})_{ij}$. Thus, the set of $W^\star$ maximizing equation 38 is in one-to-one correspondence with the set of $W_{\mathrm{p}}^\star$ maximizing the same quantity (given a fixed $W_{\mathrm{s}}$). Conditional on a given graph $\mathcal{G}_m$, and given any matrix $W^\star$ picked out by the supremum over $W|\mathcal{G}_m$, there exists a matrix $W_{\mathrm{p}}$ for which our score function, equation 38, is maximized with $W^\star = W_{\mathrm{s}}^\star \circ W_{\mathrm{p}}^\star$.

This allows us to extend Theorem 3.2 from [18], with an additional condition on $W_{\mathrm{s}}$:

**Theorem A.1.** *For a graph $\hat{\mathcal{G}} \in \mathcal{D}$, where $\mathcal{D} \subset DAG$, if $\hat{\mathcal{G}} \in argmax_{\mathcal{G} \in DAG}\mathcal{S}_{\mathcal{I}}(\mathcal{G})$ with $\mathcal{S}_{\mathcal{I}}$ defined in equation 38, and furthermore if (i) the density model has sufficient capacity to exactly represent the ground truth distributions, (ii) a given set of interventions $\mathcal{I}$ satisfies $\mathcal{I}$-faithfulness for the true graph and distributions $(\mathcal{G}^\star, P_X)$, (iii) the density models are strictly positive, (v) the ground truth densities $p^{(k)}(\mathbf{X})$ have finite differential entropy, and (vi) the elements of $W_{\mathrm{s}}$ are nonzero wherever the elements of the adjacency matrix of the true graph $\mathcal{G}^\star$ are nonzero, then $\hat{\mathcal{G}}$ is $(\mathcal{I}, \mathcal{D})$-Markov equivalent to $\mathcal{G}^\star$, for small enough $\eta$.*

In practice, condition (vi) can be supported or enforced by adding an additional regularization term or hard constraint to prevent $W_S$ from becoming too sparse.

*Remark* A.2. . Since learning $W_s$ in the upper learning is conducted with any constraints (in particular without an acyclicity constraint), the upper-level optimization is the same in any $\mathcal{L}_1$-regularized regression problem. Hence it is easy for the assumption on the nonzero elements of $W_{\mathrm{s}}$ to be satisfied.

# B  Data Appendix

## B.1  Dynamic DAG Learning and Data Normalization

Recently [58] have found there may be biases existing in the data generating process, and data normalization is suggested. Here we would like to point out that time series DAG learning differs from the i.i.d. setting, and it is not obvious how the conclusion from [58] is directly applicable to time series data and what sort of standardization even makes sense to generate data in our setting. For example, standardization in i.i.d. data is across samples, which is different from typical time series standardization across time. Moreover, time series data like Brownian motion naturally increases variance over time, so real-world datasets should indeed show higher variance for variables at later epochs. Yet another complexity is that the data in our setting is non-stationary. It would be an interesting research question to investigate how standardization of data might affect time series DAG learning in general.

## B.2  Implementation Details for Synthetic Datasets

We use the following steps and hyper-parameters to generate the data.

- For step 1), we use the Erdos-Renyi model to generate shared intra-slice graph $\boldsymbol{W}_s^a$ with degree of 3 for a total $d$ nodes, from 5 to 20, with its weights sampled uniformly at random from $\mathcal{U}([-2.0, -0.5] \cup [0.5, 2.0])$. We use the same Erdos-Renyi model to generate inter-slice graph $\boldsymbol{W}_s^b$ with degree of 3, with its weights sampled uniformly at random from $\mathcal{U}([-0.5, -0.3] \cup [0.3, 0.5])$. Then for each domain $m$, we randomly remove or add 2 edges from the previous domain (randomly generated in the first domain), to ensure temporal smoothness. Then we generate weights of $\boldsymbol{W}_{\mathrm{p},m}$ similarly as $\boldsymbol{W}_s$, but with edge weight ranges of 5 times smaller, i.e., $\mathcal{U}([-0.4, -0.1] \cup [0.1, 0.4])$ for $\boldsymbol{W}_p^b$ and $\mathcal{U}([-0.1, -0.06] \cup [0.1, 0.06])$ for $\boldsymbol{W}_p^a$.
- For step 2), we generate data with the first autoregressive order, where data only depends on the previous time slice. The underlying function between a node and its parents becomes a two-layer MLP with a sigmoid activation function. In the MLP setting, the weights of the MLPs are sampled uniformly from $\mathcal{U}([-2.0, -0.5] \cup [0.5, 2.0])$, and then weights in the first layer are updated by the parental weights from $\boldsymbol{W}^a$ and $\boldsymbol{W}^b$ structures. The number of hidden neuron sizes is set to be 10. We generate 10 sequences with 100 time slices each with standard Gaussian noises. We estimate one graph from each sequence and report the average SHD along with its standard deviation.
- For step 3), to generate interventional data, we flip a fair coin to decide whether to intervene at any time slice, in which case we sample one node uniformly to be the intervened node. We sample intervened nodes' values from another distribution. We assume every node has a probability of 0.1 to be intervened upon at each time slice, and if a node is chosen, the soft intervention comes from a different 2-layer MLP, depending on the values of its parent nodes per its graph and similarly generated as the first MLP. This soft intervention distribution can be very different from the observation distribution.

Hence, overall we generate 10 sequences for 10 domains, each sequence with 100 time steps. For the hyperparameter values, we use a separate validation dataset to choose the best performing hyperparameters for each method per SHD[2]. We search for the best value of each of 5 parameters sequentially, including two $\mathcal{L}_1$ penalty coefficients for $\boldsymbol{W}^a$ and $\boldsymbol{W}^b$, the threshold to obtain final $\boldsymbol{W}^a$ and $\boldsymbol{W}^b$, and the hidden neuron size. For $\lambda_a$ and $\lambda_w$, we search over a value range of $\{10^{-5}, 10^{-4}, 10^{-3}, 10^{-2}, 10^{-1}\}$. Graph threshold search range is set to be $\{0.001, 0.01, 0.05, 0.1, 0.2, 0.3\}$, and neuron size range is searched over $\{8, 16, 32, 64\}$. We use a two-layer MLP architecture in the experiments: $\tilde{\boldsymbol{X}}_i = W_1 \sigma((\boldsymbol{W}_{\mathrm{p},i}(\boldsymbol{W}_{\mathrm{s}}) \circ \boldsymbol{W}_{\mathrm{s}})\boldsymbol{X}_i))$, where $\boldsymbol{W}_1$ is the MLP layer weight and $\sigma$ is the ReLU activation function. For other baseline methods, we use the default parameter settings.

## B.3  Implementation Details for Sprite World Datasets

To generate a meta-learning dataset, we first generated a random Erdos-Reny graph, replicated it to the next step, and added a single edge for each variable at two consecutive time steps to introduce

---

[2]Alternatively, if datasets do not contain ground truth graphs, we can use the reconstruction loss on data as an alternative to measuring the quality of the learned graph

correlation across time steps. When we switch to a different domain, we randomly add or remove an edge in the instantaneous graph and time-delayed graph. Since we are interested in structure learning, we used a random policy agent that randomly selects at most two objects and intervene the value with a truncated normal distribution at each time step. The details of hyper-parameters are given as follows. For non-linear SEM, we utilized a two-layer MLP with the weight in the first layer ranges between $(0.5, 2.0)$ or $(-2.0, -0.5)$ and the second layer between $(0.01, 0.5)$ or $(-0.5, -0.01)$ and the bias follows the Gaussian distribution with scale 1. We again generate 10 domains with 10 different sequences with 100 time steps, and report the SHD performances across sequences.

## B.4 Ablation Study: Linearity and Private Parameters

We perform an ablation study with two new baselines, 1) a multi-task DAG learning where only shared parameters are learned (named MT-DAG), and 2) a linear meta-learning formulation (named Meta-D2AG-Linear), on synthetic datasets with scale-free (SF) graphs.

The data generation process is the same as ER graphs as discussed, except for the graph types. Our Meta-D2AG shows the best performance against these two variants of the proposed approaches, as well as the best performing IDYNO-Soft From Section 4.1, in both batch and online settings.

| Var. Size | IDYNO | MT-DAG | Meta-D2AG-Linear | Meta-D2AG |
|---|---|---|---|---|
| $d = 5$ | 11±3.9 | 9.3±0.9 | 21±0.5 | **7.2±1.4** |
| $d = 10$ | 23±3.1 | 25±4.4 | 37±1.7 | **18±0.8** |
| $d = 15$ | 51±3.2 | 58±15 | 50±1.8 | **24±2.6** |
| $d = 20$ | 42±5.3 | 46±9 | 67±4.1 | **35±1.9** |

Table 2: SHD Results for Scale-Free Graph (SF) in the Batch Setting

| Var. Size | IDYNO | MT-DAG | Meta-D2AG-Linear | Meta-D2AG |
|---|---|---|---|---|
| $d = 5$ | 8.8±0.3 | 24±1.6 | 21±1.3 | **8.7±0.8** |
| $d = 10$ | 29±1.9 | 37±1.6 | 46±2.7 | **18±0.7** |
| $d = 15$ | 54±2.9 | 55±4.9 | 53±2.4 | **27±0.4** |
| $d = 20$ | 47±5.5 | 77±6.1 | 66±5.8 | **36±1.2** |

Table 3: SHD Results for Scale-Free Graph (SF) in the Online Setting

## B.5 Ablation Study: Sequence Length

We conduct an experiment on the effect of sample sizes vs SHD. We use variable dimension $d = 20$ and 10 total sequences, and the rest setting are the samw as in Section 5.1. When the sequence lengths are lower, all methods perform worse, but our method remain competitive.

| Sequence Length | 10 | 50 | 100 |
|---|---|---|---|
| GOLEM | $115 \pm 20$ | $102 \pm 10$ | $81 \pm 10$ |
| DYNOTEAR | $187 \pm 23$ | $150 \pm 20$ | $89 \pm 16$ |
| IDYNO | $129 \pm 16$ | $97 \pm 14$ | $80 \pm 21$ |
| JPCMCI+ | $132 \pm 21$ | $130 \pm 20$ | $119 \pm 9$ |
| NTS-NOTEARS | $119 \pm 15$ | $95 \pm 5$ | $63 \pm 13$ |
| Meta-D2AG | $101 \pm 10$ | $80 \pm 7$ | $46 \pm 6$ |

Table 4: Performance across different sequence lengths in Online Setting, $d = 20$.

## B.6 Ablation Study: Random Variance

To study the impact of variable variance, we repeat the synthetic dataset in Experiment 5.1 with random noises between 0.1 and 2 for each variable noise term, with 4 different variable sizes. Below are the results against baselines. The relative errors are higher than of fixed variance of 1, but the Meta-D2AG method remains the best among methods tested.

| d | 5 | 10 | 15 | 20 |
|---|---|---|---|---|
| GOLEM | $5.9 \pm 0.9$ | $26.6 \pm 1.8$ | $43.7 \pm 7.8$ | $93.5 \pm 7.9$ |
| DYNOTEAR | $6.7 \pm 1.1$ | $22.3 \pm 3.0$ | $60.4 \pm 4.6$ | $95.6 \pm 5.3$ |
| IDYNO | $6.5 \pm 1.1$ | $26.5 \pm 1.1$ | $57.5 \pm 5.7$ | $90 \pm 9.2$ |
| JPCMCI+ | $6.6 \pm 0.1$ | $23.6 \pm 1.1$ | $85 \pm 10.2$ | $105 \pm 19.8$ |
| NTS-NOTEARS | $12.3 \pm 0.5$ | $33.3 \pm 2.1$ | $55.7 \pm 5.3$ | $104 \pm 10.2$ |
| Meta-D2AG | $6.6 \pm 0.1$ | $22.9 \pm 2.2$ | $46.7 \pm 4.6$ | $82.7 \pm 8.9$ |

Table 5: SHD with random variable variance across varying dimensions $d$ in the online setting.

