# OpenReview forum: "Meta-D2AG: Causal Graph Learning with Interventional Dynamic Data"
_NeurIPS.cc/2025/Conference — NeurIPS 2025 poster_

### Official Review · Reviewer_gc1P · 2025-06-30

**Clarity:** 3
**Significance:** 3
**Originality:** 3
**Rating:** 4
**Confidence:** 4

**Summary:**

This paper addresses causal discovery in dynamic, non-stationary time series, proposing Meta-D²AG—a meta-learning framework based on bilevel optimization that models both parameter and structural changes as domain shifts prompted by interventions. Unlike prior approaches, Meta-D²AG explicitly treats distribution shifts as domains and leverages both historical data and online learning to enable rapid adaptation with few samples, using a first-order optimization method for efficiency. Theoretical guarantees on identifiability and convergence are provided. Empirical results on synthetic data and a reinforcement learning environment show that the method outperforms recent baselines in terms of structural Hamming distance (SHD), under both batch and online evaluation.

**Questions:**

- The synthetic time-series experiments in the paper only consider non-linear structural equation models. To better assess the generality of the proposed method, it would be important to include experiments on linear vector autoregressive (VAR) models.
- The work would benefit from comparisons with additional baselines to better demonstrate the performance advantage, such as NTS-NOTEARS[1] and NGC[2].
- The authors mention using default hyperparameter settings for other methods (Line 1002). To ensure a fair comparison, it is important to conduct some level of hyperparameter tuning for these baselines.
- The paper should report runtime comparisons between Meta-D²AG and other methods to clarify whether the performance gain comes at the cost of increased computational overhead.
- The experiments are conducted under a fixed sample size. Evaluating performance across varying sample sizes would be helpful to understand the robustness and scalability of the method under different data regimes.
- The method assumes that domain boundaries (e.g., change points) are predefined, which may limit its applicability to real-world time series with unknown or unobserved distribution shifts. Could the authors discuss how Meta-D²AG might be extended to incorporate automatic domain segmentation or change-point detection techniques?

Reference
[1] Sun X, Schulte O, Liu G, et al. Nts-notears: Learning nonparametric dbns with prior knowledge[J]. arXiv preprint arXiv:2109.04286, 2021.
[2] Tank A, Covert I, Foti N, et al. Neural granger causality[J]. IEEE Transactions on Pattern Analysis and Machine Intelligence, 2021, 44(8): 4267-4279.

**Ethical Concerns:**

["NO or VERY MINOR ethics concerns only"]

**Limitations:**

Nothing

**Paper Formatting Concerns:**

Nothing

**Quality:**

3

**Strengths And Weaknesses:**

### Strengths
- Addresses a Practically Significant Problem: The challenge of learning causal structure from non-stationary, interventional time series is of high interest in machine learning, with broad applications.
- Clear Framing and Motivation: The paper frames the problem well, connecting real-world distribution shifts with the technical approach; the definition of "domains" via interventions is clearly articulated in Section 1.
- Methodological Contribution: Introduces a new bilevel meta-learning approach enabling separation and adaptation of shared versus domain-specific parameters (Section 3.2, “Meta-Learning for Dynamic DAG”).
- Online and Efficient Learning: Offers a mechanism for online adaptation using a sliding window and efficient penalty-based bilevel gradient descent (Section 3.4); this is concrete and distinct from prior work.
- Solid Theoretical Foundation: Provides theoretical convergence guarantees (Theorem 4.1, Section 4), with clear statement of assumptions. The regret bound and discussion around use of sliding window optimization are appropriate for the task.
- Empirical Validation: The experiments compare against a broad set of contemporary baselines, covering both synthetic and RL datasets. Quantitative evidence—Strong empirical results are shown in Figure 1a and 1b (showing that Meta-D²AG achieves consistently lower SHD than competitors as variable numbers/d increases). Table 1 shows strong SHD performance in the Sprite World environment, with scores better than all baselines for both 5 and 10 objects.
- Clarity of Exposition: Most sections are clearly written and logically organized; the relationship between conceptual formulation and empirical outcomes is well maintained.
- Explicit Limitation Statement: Section 6 makes limitations explicit, notably the use of predefined domains, and discusses societal impact.

### Weaknesses
- Empirical Scope Is Limited: The main experimental validation focuses on synthetic datasets generated via SEMs and a simulated environment (Sprites World), lacking real-world datasets. This limits the demonstrated utility of the approach and may not fully capture practical challenges (Section 5).
- Assumption of Known Domain Boundaries: The method assumes access to temporally segmented domains (Section 3), but in practice, detecting or segmenting domains (especially under gradual or unknown shifts) is non-trivial. This is acknowledged in Section 6, but the lack of any attempt to address it (even with heuristics) restricts applicability and significance.
- Comparison Baselines and Fairness: The comparison in Section 5 does not fully leverage methods that are specifically tailored to online or meta-learning in dynamic graphs. Baselines appear limited to those with available code, potentially missing very recent or specialized meta-learning baselines. Furthermore, for online settings, all data up to the current domain is used for baselines, but it is not clear whether this is fully fair for all compared methods.
- Reproducibility and Implementation Details: The experiments lack sufficient implementation details in the main paper (Section 5).
- Hyperparameter settings, architectures, loss settings, and optimization strategies (e.g., window size, regularizer values) are only mentioned briefly or deferred to the appendix, affecting reproducibility and clarity.
- Interpretability of Quantitative Gains: While SHD is a standard metric, it is not clear how sensitive the methodology is to parameter choices. Ablation and sensitivity analyses are reported as being in the appendix, so the main paper does not allow assessment of robustness (Section 5).
- Computational Complexity Discussion Lacks Details: While complexity is discussed in Section 4, there is no empirical measurement or comparison of runtime/efficiency, especially as the window size or number of domains grows.

---

> ### Author Rebuttal · Authors · 2025-07-31
>
> **Sample size v.s. performance**: Per request, we conducted following experiment on samples sizes vs SHD. We use variable dimension $d=20$ and $10$ total sequences. When sample is low, all methods perform worse, but our method remain competitive.
>
> | sequence length | 10  | 50 | 100|
> |---|---|----|----|
> | GOLEM  | $115\pm20$  |  $102\pm10$     |  $81\pm 10$   |
> | DYNOTEAR  |  $187\pm23$    |   $150\pm20$  | $89\pm16$   |
> | IDYNO  |  $129\pm16$   | $97\pm14$    |   $80\pm21$   |
> | JPCMCI+ | $132\pm21$     |   $130\pm20$  | $119\pm9$   |
> | NTS-NOTEARS   | $119\pm15$    |   $95\pm 5$   | $63\pm13$   |
> | Meta-D2AG   | $101\pm 10$   |  $80\pm 7$    | $46 \pm 6$   |
>
>
> **Run time**: We include the running time of each method below, for the experiment with random variance (Per Reviewer mYVS's quest).
>
> | d | 5 | 10 | 15 | 20 |
> |---|---|----|----|----|
> | GOLEM  | $139\pm50$  |  $141\pm12$  |   $241\pm11$  |  $420 \pm 10$  |
> | DYNOTEAR  |  $1.4\pm0.1$  |  $2.9\pm2.1$  |  $5.9\pm3.3$ | $21\pm5.3$   |
> | NTS-NOTEARS | $247\pm 66$   | $170\pm2$   |   $377 \pm 33$ | $480\pm29$   |
> | Meta-D2AG  | $232\pm20$  | $265\pm 30$   |  $260\pm 4.6$ | $350 \pm 8.9$   |
>
>
> **Assumption of Known Domain Boundaries: The method assumes access to temporally segmented domains (Section 3), but in practice, detecting or segmenting domains (especially under gradual or unknown shifts) is non-trivial**: Thank you for the comment. Distribution shift detection is an active area of research, and we can adopt various strategies, such as measuring prediction error, using distribution distance, and/or hypothesis testing to detect shift [1] and form a new domain. We will add such a discussion.
>
> [1] Lu et al, "Learning under Concept Drift: A Review", IEEE Transactions on Knowledge and Data Engineering.
>
>
>
> **Other comments**: We thank the reviewer for all the comments. Our reported results uses the general default hyperparameter settings, but during the experimental study, we conducted some hyperparameter search and find the default settings are generally robust and hence we choose to use them. We did not have time to finish linear experiments, but our method focuses on nonlinear settings due to more flexibility and modeling power. Per Suggestion, we also included NTS-NOTEARS results here. We will release codes.

---

### Official Review · Reviewer_fhwy · 2025-07-02

**Clarity:** 3
**Significance:** 3
**Originality:** 3
**Rating:** 4
**Confidence:** 2

**Summary:**

This paper investigates causal discovery by proposes an online meta-learning based method to learn dynamic DAG structures from potentially nonlinear and non-stationary time series datasets. Different from existing work, this paper explicitly treats data collected at different time points with distribution shifts as distinct domains, which is assumed to occur as a result of external interventions. The authors design a first-order optimization approach to solve the meta-learning framework, and provide a detailed theoretical analysis to establish the identifiability conditions and the convergence of the learning process. Extensive experiments are conducted on benchmark datasets to demonstrate the promising performance of the proposed approach.

**Questions:**

N.A.

**Ethical Concerns:**

["NO or VERY MINOR ethics concerns only"]

**Final Justification:**

I’ll retain my score

**Limitations:**

yes

**Paper Formatting Concerns:**

N.A.

**Quality:**

3

**Strengths And Weaknesses:**

Strengths

S1. The proposed meta-learning framework can handle both observational and interventional data distribution with nonlinear relationships by formulating a non-stationary collection of dynamic datasets as different domains.

S2. A theoretical analysis is provided to establish the identifiability conditions in the batch setting and provides convergence rates in the online setting.

S3. Experimental results on benchmark datasets demonstrate the superiority of the proposed approach.

Weaknesses

W1. The authors only use SHD as the metric in the experimental study. Maybe more metrics like TPR could be used to report the performance of the proposed approach.

W2. In Figure 1b, is IDYNO-Soft plotted with error bars? Why the other methods do not have error bars?

W3. It would be easier for readers to understand the general idea if the authors could use some figure to show the overall framework.

---

> ### Author Rebuttal · Authors · 2025-07-31
>
> **Metrics**: We use SHD as it is the standard metric. We will add more results using TPR, but for the Sprites World  dataset in Section 5.2 for example, our method has TPR = 0.73 while the other baselines are in range of 0.5 to 0.65.
>
> **Error bars**: the errors bars for some methods are not rendered correctly, and we will fix them. In general they are small related to the mean, as seen in Table 1, 2, and 3.
>
> **Overall figure**: thank you for the suggestion, and we will add an overview figure to highlight our methods, complementary to our description in Section 3.2 and 3.3. It will consists of two sub-figures, each for offline and online versions of our methods. Each sub-figure will contain both meta-train and meta-test phrases. For meta-train, we will highlight different domain data $\bf{X}$, which parameters are shared and private ($\Phi_s$ and $\Phi_{p}$'s), learned graphs $A_{m} = W_s \circ W_{p,m}$ for each domain $m$, and its joint loss function. For meta-test, we will highlight how meta parameters are fixed and only need to solve for new $\Phi_{p,m}$ for a new domain.

---

> > ### Comment · Reviewer_fhwy · 2025-08-08
> >
> > Thank you for your response. As I’m not an expert in this area, I’ll retain my positive score.

---

### Official Review · Reviewer_mYVS · 2025-07-03

**Clarity:** 2
**Significance:** 2
**Originality:** 3
**Rating:** 5
**Confidence:** 2

**Summary:**

The paper proposes a new dynamic DAG discovery algorithm, Meta-DAG, based on online meta-learning. It can learn from potentially nonlinear and non-stationary time series. Meta-DAG involves a new online meta-learning framework to take advantage of the temporal transition among existing domains such that it can quickly adapt to new domains with few measurements. The authors also provide theoretical analysis of the convergence of the algorithm as well as identifiability results. Meta-DAG shows improved performance on the conducted tests.

**Questions:**

1. One of the main claims made in the introduction section was the ability of the proposed method to quickly adapt to new data distribution and achieve higher graph learning accuracy with a few samples. However, this claim was never experimentally validated. It would help to compare the baselines and the proposed method on the sample requirements to learn the parameters on new domain data.
2. As detailed in Appendix B.2, the noise variance seems to be set to 1 for all the variables. What happens when the noise variance is not constant and is random? Does Meta-DAG exhibit the same trend event in this case?
3. The conversion of (6a), (6b) to (7) seems a bit unclear. It would help to have a more elaborate discussion on this in the main paper or the appendix.

If the authors can satisfactorily address the questions above, I would be inclined to raise my overall recommendation.

(Minor comment) There's a typo in line 220.

**Ethical Concerns:**

["NO or VERY MINOR ethics concerns only"]

**Final Justification:**

The new experiments provided during the rebuttal addresses my concerns.

**Limitations:**

Yes

**Paper Formatting Concerns:**

Yes

**Quality:**

2

**Strengths And Weaknesses:**

Quality: The paper is technically sound with good theoretical results. The experimental validation showcases improved performance over the baseline. However, one of the main claims of the paper, Meta-DAG can "quickly adapt to a new data distribution and achieve higher graph learning accuracy with a few samples" hasn't been shown experimentally.

Clarity: The presentation is sound. However, the discussion in section 3.4 could benefit from further explanation. Especially around eq (7).

Significance: The proposed work addresses the difficult task of dynamic causal discovery from multiple domains. The experimental results show that the proposed method Meta-DAG outperforms the baselines in both synthetic and real-world experiments. However, it would help to have addition experiments testing the limits of the proposed method, see Questions.

Originality: The proposed work is novel and it introduces an online meta-learning based procedure for dynamic causal graph learning.

---

> ### Author Rebuttal · Authors · 2025-07-31
>
> **Sample size v.s. performance**: Per request, we conducted following experiment on samples sizes vs SHD. We use variable dimension $d=20$ and $10$ total sequences. When sample is low, all methods perform worse, but our method remain competitive.
>
> | sequence length | 10  | 50 | 100|
> |---|---|----|----|
> | GOLEM  | $115\pm20$  |  $102\pm10$     |  $81\pm 10$   |
> | DYNOTEAR  |  $187\pm23$    |   $150\pm20$  | $89\pm16$   |
> | IDYNO  |  $129\pm16$   | $97\pm14$    |   $80\pm21$   |
> | JPCMCI+ | $132\pm21$     |   $130\pm20$  | $119\pm9$   |
> | NTS-NOTEARS   | $119\pm15$    |   $95\pm 5$   | $63\pm13$   |
> | Meta-D2AG   | $101\pm 10$   |  $80\pm 7$    | $46 \pm 6$   |
>
>
>
> **Random variance**: Per suggestion, we repeat the synthetic dataset in Experiment 1 with random noises between 0.1 and 2 for each variable noise term, with 4 different variable sizes $d=5, 10, 15,$ and $20$. Below are the results against baselines. The relative errors are higher than of fixed variance of 1, but the Meta-D2AG method remains the best among methods tested.
>
> | d | 5 | 10 | 15 | 20 |
> |---|---|----|----|----|
> | GOLEM  | $5.9\pm0.9$  |  $26.6\pm1.8$  |  $43.7\pm7.8$    |  $93.5 \pm 7.9$  |
> | DYNOTEAR  |  $6.7\pm1.1$  |  $22.3\pm3.0$  |   $60.4\pm4.6$ | $95.6\pm5.3$   |
> | IDYNO  |  $6.5\pm1.1$ | $26.5\pm1.1$   | $57.5\pm5.7$   |   $90\pm9.2$ |
> |  JPCMCI+ | $6.6\pm0.1$   | $23.6\pm1.1$   |   $85.3\pm10.2$ | $105\pm19.8$   |
> | NTS-NOTEARS | $12.3\pm0.5$   | $33.3\pm2.1$   |   $55.7\pm 5.3$ | $104\pm10.2$   |
> | Meta-D2AG  | $6.6\pm0.1$  | $22.9\pm 2.2$   |  $36.7\pm 4.6$ | $82.7 \pm 8.9$   |
>
>
>
> **The conversion of (6a), (6b) to (7) seems a bit unclear**: We move both the objective and constraints in the inner problem as different penalty terms to the outer problem, and thus form as a single objective. Thank you for your comments, and we will clarify this and other typos in the paper.

---

> > ### Comment · Reviewer_mYVS · 2025-08-05
> >
> > Thank you for your response and clarifications. I have raised my score.

---

### Official Review · Reviewer_ufRx · 2025-07-08

**Clarity:** 1
**Significance:** 3
**Originality:** 3
**Rating:** 4
**Confidence:** 1

**Summary:**

The authors propose Meta-D²AG, a dynamic DAG discovery algorithm for time series data, which is based on meta-learning. The model adapts to non-stationary distributions over time by considering different time windows as separate domains and uses meta-learning to learn shared parameters across domains while quickly adapting to new domains with few measurements.

**Questions:**

1. Could the authors provide a visual illustration or diagram of the overall model pipeline to help non-expert readers better understand the approach?

2. Have the authors evaluated the model on any real-world datasets, and if not, can they comment on its expected performance and applicability beyond synthetic settings?

3. Given that the method relies on predefined domain splits, have the authors considered ways to automatically detect distribution shifts or segment domains in practice when such labels are not available?

**Ethical Concerns:**

["NO or VERY MINOR ethics concerns only"]

**Final Justification:**

The authors' responses addressed my main concerns, particularly around model clarity and real-world applicability. Combined with the positive feedback from other reviewers, this led me to raise my score, despite some remaining difficulty in fully assessing the technical details.

**Limitations:**

yes, limitation were adressed in the conclusion

**Paper Formatting Concerns:**

no concerns

**Quality:**

2

**Strengths And Weaknesses:**

### Weaknesses
- The paper could benefit from clearer examples or visualizations to help readers intuitively understand how the method works
- The proposed approach is technically complex and may be difficult for non-experts to understand or implement without significant background knowledge
- Most experiments are done on synthetic or simulated data. It's unclear how the method performs on real-world datasets.

### Strengths
- The results show significant improvements over baseline methods
- The paper addresses an important problem innovatively by combining methods from different research areas
- The method is flexible and works with observational and interventional data

---

> ### Author Rebuttal · Authors · 2025-07-31
>
> **Visual illustration or diagram of the overall model pipeline to help non-expert readers better understand the approach** : thank you for the suggestion, and we will add an overview figure to highlight our methods, complementary to our description in Section 3.2 and 3.3. It will consists of two sub-figures, each for offline and online versions of our methods. Each sub-figure will contain both meta-train and meta-test phrases. For meta-train, we will highlight different domain data $\bf{X}$, which parameters are shared and private ($\Phi_s$ and $\Phi_{p}$'s), learned graphs $A_{m} = W_s \circ W_{p,m}$ for each domain $m$, and its joint loss function. For meta-test, we will highlight how meta parameters are fixed and only need to solve for new $\Phi_{p,m}$ for a new domain.
>
> **Have the authors evaluated the model on any real-world datasets, and if not, can they comment on its expected performance and applicability beyond synthetic settings**: Thank you for the comment. While we do not have public datasets with ground truth causal graphs to test our algorithm on, we emphasize that our algorithm should be same or better performance than existing methods, as our framework includes existing methods as inner problem. Our methods are well suited to real world datasets where stationarity is violated.
>
> **Given that the method relies on predefined domain splits, have the authors considered ways to automatically detect distribution shifts or segment domains in practice when such labels are not available** : Thank you for the comment. Distribution shift detection is an active area of research, and we can adopt various strategies, such as measuring prediction error, using distribution distance, and/or hypothesis testing to detect shift [1] and form a new domain. We will add such a discussion.
>
> [1] Lu et al, "Learning under Concept Drift: A Review", IEEE Transactions on Knowledge and Data Engineering.

---

> > ### Comment · Reviewer_ufRx · 2025-08-06
> >
> > Thank you for the detailed and thoughtful responses to my questions.
> > The planned addition of a visual overview is much appreciated and will certainly help improve the accessibility of the method. Your clarifications regarding applicability to real-world, non-stationary datasets, as well as your discussion on potential strategies for automatic domain segmentation, address my concerns to a reasonable extent.
> >
> > While I may not fully grasp all of the technical details, I recognize the contribution this work makes—particularly when considered alongside the strong evaluations from the other reviewers. In light of your clarifications and the broader consensus, I will raise my score.

---

### Decision · Program_Chairs · 2025-09-17

**Decision:**

Accept (poster)

**Comment:**

This paper introduces a novel framework for causal discovery in dynamic and non-stationary settings, combining methodological innovation with solid theoretical grounding and empirical evidence. The reviewers recognize the importance of the problem, the originality of the approach, and the overall soundness of the work. While some limitations were noted in clarity, scope of evaluation, and assumptions, the authors have provided satisfactory clarifications and additional results during the rebuttal. Overall, the strengths of the paper clearly outweigh the weaknesses, and I recommend acceptance.